# Fabrication of Mupirocin-Loaded PEGylated Chitosan Nanoparticulate Films for Enhanced Wound Healing

**DOI:** 10.3390/ijms25179188

**Published:** 2024-08-24

**Authors:** Shajahan Azeez, Anbazhagan Sathiyaseelan, Kaviyarasan Venkatesan, Myeong-Hyeon Wang

**Affiliations:** 1Centre for Advanced Studies in Botany, University of Madras, Guindy Campus, Chennai 600025, India; manikavi53@gmail.com; 2Department of Medical Biotechnology, Aarupadai Veedu Medical College and Hospital, Vinayaka Mission’s Research Foundation (DU), Puducherry Campus, Puducherry 607402, India; 3Department of Bio-Health Convergence, Kangwon National University, Chuncheon 24341, Republic of Korea; sathiyaseelan.bio@gmail.com

**Keywords:** chitosan, mupirocin, PEG, composite film, in vivo, wound healing

## Abstract

Chitosan-based biomaterials are being investigated for their unique properties that support skin regeneration and wound healing. This study focused on the preparation and characterization of a mupirocin (Mup)-loaded PEGylated chitosan (CS-PEG) nanoparticulate film (NF) [CBNF]. The CBNF was characterized using FTIR spectroscopy and SEM analysis. The results demonstrated that CBNF was successfully incorporated into the composites, as shown by functional group modification through FTIR analysis. Additionally, the SEM micrograph revealed the deposition of nanoparticles (<200 nm) on the surface of transparent CBNF. The film has higher water absorption (≥1700%) and moderate water retention ability within 6 h. Furthermore, histological findings showed significant development, with re-epithelialization and granulation of tissues after 19 days, indicating the healing efficiency of CNBF. These results suggest that drug-loaded films could be an effective carrier and delivery agent for Mup-like anti-inflammatory drugs.

## 1. Introduction

Wound healing is the body’s natural process of regenerating dermal and epidermal tissue. The healing process is crucial to protect the wound against bacterial infection, especially considering that the warm and moist environment of wound beds provides an ideal setting for bacteria’s rapid growth and proliferation. These bacterial colonies swiftly generate an extracellular polymer matrix on the wound surface, enhancing their resistance against the host immune system [1]. Unhealed wounds bring about significant discomfort and vulnerability. The ability of the skin to repair itself after a minor wound is remarkable, but when the damage is severe or occurs in large areas of skin, proper and immediate coverage of the wound surface with an adequate dressing is needed to protect the wound and to accelerate wound healing. The first and critical phase, hemostasis, necessitates prompt cessation of bleeding [2,3]. Successful wound healing involves eradicating infections and facilitating subsequent cell growth and tissue restructuring. Wound repairing is a complex process involving an integrated response by many different cell types and growth factors to rapidly restore skin integrity and protective function after injury [4]. When the body’s internal repair mechanisms, such as cell growth and tissue remodeling, fall short, the external application of wound-healing treatments becomes effective. Thus, it becomes imperative to instill wound dressings with exceptional antibacterial capabilities and efficient drug delivery mechanisms to promote optimal wound healing [5,6]. Understanding these mechanisms allows for developing advanced wound dressings that facilitate and expedite healing.

Wound dressings are vital for protecting the wound bed and creating an environment conducive to healing. Effectively managing wound exudate within this environment is essential for tissue repair [2]. Excessive exudate can impede cell growth and disrupt healing, potentially fostering bacterial and pathogenic growth [7,8]. Healing dermal wounds with macromolecular agents such as natural polymers is preferred to skin substitutes owing to many advantages such as their biocompatibility, nonirritant and non-toxic properties, and ease and safety of application on the dermis [9]. Biomaterials are natural polymers and are biodegradable, used in regenerative medicine, implantable materials, controlled-release carriers, or scaffolds for tissue engineering. Cellulose, chitin, chitosan, and gelatin are widely used natural polymers. When used as drug delivery carriers, natural polymers are degraded into biologically accepted compounds, often through hydrolysis, which leaves the incorporated medications behind [10]. Biodegradable polysaccharide chitosan has a history of use as a vehicle in pharmaceutical preparations [11]. Numerous biomaterials have been examined for remedies for skin injuries. Despite various commercial dressing options for treating these lesions, many products impose a significant financial burden on patients [6]. Moreover, the ideal dressing should fulfill requirements for swift wound healing and possess suitable mechanical and antimicrobial properties, alongside favorable compatibility within the body, among other sought-after characteristics [12].

Chitosan (CS) is a linear polysaccharide structure comprising two monomers: 2-amino-2-deoxy-β-D-glucopyranose (D-glucosamine) and 2-acetamido-2-deoxy-β-D-glucopyranose (N-acetyl-D-glucosamine) [13]. This biopolymer exhibits biocompatibility, biodegradability, and non-toxicity. Numerous findings indicate the beneficial use of chitosan in gels [14,15,16], films [17,18], and meshes for treating skin wounds [16], demonstrating positive outcomes in both human and animal studies [19,20]. CS is a promising biomaterial in the pharmaceutical field, particularly as a novel drug carrier. Its widespread use as a cationic bio-polysaccharide in delivering therapeutic agents aims to prevent and/or manage bacterial infections while enhancing healing [21,22]. Chitosan interacts with bacteria’s surface charge to increase tissue resistance by accumulating macromolecules. The bactericidal effect of the polymer is inherent due to its protonated amino group [22]. Embedding the medications within the film structure creates films containing antibacterial and antibiotic drugs. These films serve as a shield against microorganisms, effectively averting secondary infections. It creates an optimal environment that promotes the wound-healing process, making chitosan an ideal biopolymer for wound healing and tissue regeneration.

To overcome some of the difficulties associated with many topical preparations used to control wound infections, mupirocin—an effective treatment for secondarily infected, minor wounds with better tolerance—is the drug of choice for preventing and treating bacterial infections. This study focuses on producing a nanoparticle-based film derived from fungal chitosan along with PEG and mupirocin to provide wound-healing properties and a more precise and effective route of administration than simple topical application (Figure 1). This investigation is a continuation of our recent work [13] involving the administration of mupirocin-loaded chitosan-based nanoparticle films to wounds in Wistar albino rats, and the results are analyzed in conjunction with the relevant literature.

## 2. Results

### 2.1. Fourier-Transform Infrared Spectroscopy (FTIR) Analysis

FTIR was employed to analyze the chemical characteristics and interactions within CS, Mup, and PEG composite films (CBNF) (Figure 1). The characteristic absorption peaks of chitosan’s functional groups were observed in all blend interferograms, with varying intensities depending on the composition. CS displayed the major broad peaks at 3453 cm^−^^1^ due to the O-H stretching, and the medium peaks at 2894 cm^−^^1^ due to the C-H stretching. Further, the 1638 cm^−^^1^ and 1420 cm^−^^1^ peaks correspond to C-O and N-H stretching, respectively. In addition, the peaks at 1091 cm^−^^1^ and below are due to the C-H bending. mupirocin (Mup) exhibits the major peaks at 2930 cm^−^^1^ and 1722 cm^−^^1^, corresponding to the C-H and -C=O stretching, respectively. The peaks at 1527 cm^−^^1^, 1225 cm^−^^1^, and 1156 cm^−^^1^ are attributed to the C-O stretching. More prominent than those of chitosan, PEG’s characteristic peaks (2882, 1462, 1281, and 952 cm^−^^1^) were ascribed to C-H stretching, bending vibrations, and C-O stretching. Followingly, CS–Mup demonstrated the appearance of peaks at 2917 cm^−^^1^, 1646 cm^−^^1^, 1417 cm^−^^1^, 1104 cm^−^^1^, and 1045 cm^−^^1^. In addition, the peak intensity at 1417 cm^−^^1^ was increased from Mup, evidencing that the Mup bound with CS. Finally, the CS–PEG–Mup composite showed the characteristics peaks at 2881 cm^−^^1^, 1458 cm^−^^1^, 1354 cm^−^^1^, 1108 cm^−^^1^, 948 cm^−^^1^, and 840 cm^−^^1^. The lack of newly formed or absent peaks indicates that there was no chemical reaction between PEG and chitosan. Nonetheless, the amide I band (1648 cm^−1^) and the C-H stretching vibration (2881 cm^−1^) shifted to lower wavenumbers in the presence of CS and PEG. An attractive intermolecular interaction between chitosan and PEG is indicated by these shifts [23].

### 2.2. SEM Analysis

CBNFs were prepared as described in the Materials and Methods Section, and the image of the film’s FESEM is presented in Figure 2. By cross-observation, morphologically, the chitosan-based nanoparticulate films (CBNFs) were transparent. SEM analysis revealed a porous and interconnected structure, indicating that the CBNF has a high water-retention capacity. It is due to the high degree of deacetylation of chitosan (88%)—which can establish H-bonds with water—by which CBNF facilitates both small and macromolecules. The average size distribution of CBNF was analyzed using Dynamic Light Scattering (DLS) spectroscopy, revealing an average size of 260 nm with a polydispersity index of 0.32 (Figure 2c).

### 2.3. Swelling and Moisture Retention Abilities

A high degree of swelling helps absorb wound exudates, keeping the wound clean. Swelling ratios were measured by weight change after immersing samples in PBS (pH 7.4, room temperature) for 6 h (Figure 3a). The water absorption percentage of CBF and CBNF highly increased from their initial weight when incubating them in PBS from 1 to 6 hrs. The initial incubation at 1 h showed that 250% of the CBF and CBNF weight increased. The weight percentage then increased according to the incubation time from 250% to 1700%; however, the weight differences among the time intervals were insignificant at the fourth, fifth, and sixth hours. Moreover, the percentage of water absorption was insignificant between the samples during incubation.

In addition, the water retention ability of 2 h PBS pre-soaked CBF and CBNF was tested for 6 h at room temperature (Figure 3b). The results indicate that increasing incubation time gradually decreased the composite film weight, indicating moderate water retention ability.

### 2.4. Mupirocin Release

The in vitro Mup-release behavior from CBNF in two different pHs (5.4 and 7.4) containing medium was evaluated (Figure 3c). The results indicate that Mup release was higher at pH 5.4 over the time of incubation when compared to pH 7.4 (Figure 3c). It was indicated that the pH-responsive polymer CS facilitates the release of Mup at a higher level in pH 5.4 due to the protonation of the amine group. The 72-hour incubation of CBNF showed the cumulative Mup release at 20% and 65% for pHs 7.4 and 5.4, respectively. In addition, the Mup release at different pHs was fitted with the mathematical drug release model (Table 1). The results showed that the release of Mup at pH 7.4 (R^2^ = 0.971) and pH 5.4 (R^2^ = 0.995) highly fitted with the Higuchi drug release model. Following that, the drug release profiles in both pHs were fitted with Korsmeyer-Peppas. In addition, the release exponent (n) of Korsmeyer-Peppas for pHs 7.4 and 5.4 was 0.762 and 0.701, respectively. The exponent indicates that anomalous (non-fickian) release pattern.

### 2.5. Antibacterial Activity

The antibacterial effect of the test samples on the corresponding bacterial species is displayed in Figure 4. The experiment’s outcomes depict the test samples are efficacious against both Gram-positive and Gram-negative bacteria when correlated with mupirocin, an antibacterial control agent. Gram-negative bacteria were shown to be inhibited by CBF by 17 mm (*P. aeroginosa*), whereas it was found that Gram-positive bacteria (*S. aureus*) exhibited the least or no activity. Nonetheless, CBNF found that the maximum zone of inhibition against *P. aeroginosa* and *S. aureus* was 30 mm and 26 mm, respectively.

### 2.6. Morphological Observation of Wound Healing

In vivo experiments showed that CBNF adhered uniformly to the freshly excised wound surface [14]. The photographic image (Figure 5) showed a set of typical wound beds after creating a wound and applying CBNF. The healing patterns were observed at intervals of 5, 7, 10, 13, 17, and 19 days, and they showed that the topical application of CBNF improved wound healing. The macroscopic analysis showed that the wound beds of the animals treated with CBNF were considerably smaller when compared with those of the controls treated with chitosan PEG composite films (CBF) and mupirocin drug alone (MDA). Macroscopic findings did not reveal a significant difference in the wound contraction area until day 7. The wound area of the control animals increased during the initial period of 7 days (Figure 5), which was not observed in the animals treated with CBNF. Therefore, this observation supports the promoting role of CBNF in wound healing.

The measurement of the wound area was derived by calculating the percent area of progress from the initial wound area per day and plotting over time. The initial wound area was normalized as “0% healed,” and the area reduction progress was charted to study to the end or 100% closure. The wound plots in this study were constructed using pooled wound-healing percent area progress data. The wound-healing curves demonstrated that the application of CBNF approached 100% closure faster than the untreated control (Figure 5). During the treatment cycle, the wound area that eventually healed with CBNF was different from the slow-healing wounds. By day 19, the percent area progress of all the wound healers was recorded as follows. In the control, 83.14% closure was observed on the 19th day, while a maximum of 99.21% closure of the wound was observed in CBNF-treated wounds, followed by CBF (90.94%) and MDA (95.83%). The results showed that CBNF application showed a faster healing process by 99.21%. There remained a significant difference in percent area progress on the 19th day between fast healers and slow healers (*p* < 0.001 and <0.01, treatment and control, respectively) (Figure 6). The percentage area reduced rate of wound healing was found to increase in the following order: control < CBF < MDA < CBNF.

### 2.7. Histological Analysis of Wound Healing

Histological findings of wounded skins were studied by hematoxylin and eosin (H&E, Figure 7), and Masson’s trichome stain method (Figure 8) is in good accordance with the results. The histological examination of the CBNF treatment showed that neither specific inflammation nor reactive granulomas were observed in CBNF-applied wounds. In addition, no microorganisms were observed in the skin lesions. In the present study, the collagen deposition in the wounds was examined by Masson’s Trichome staining, as shown in Figure 8. It revealed that the wound treated with the CBNF group showed more significant collagen deposition on the 19th day as compared with that of the CBF and MDA groups. The histological section (Figure 8) of the CBNF-treated group (group IV) had compact and well-aligned collagen; however, collagen was loosely and irregularly arranged in the experimental rats of the CBF group (group II) and MDA group (group III). Re-epithelialization is essential in wound healing since the skin plays a significant barrier in protecting the host against pathogens. When the skin barrier is compromised, the immune system produces cytokines to repel invading pathogens. The risk of prolonged inflammation is dampened by accelerating re-epithelialization to close the wound and restore the skin barrier [23]. In the present study, the applied wound showed complete re-epithelialization, and a good alignment of collagen was observed.

In vivo experiments demonstrated that CBNF (chitosan-based nanocomposite film) adhered well to wounds and significantly accelerated healing compared to controls treated with chitosan PEG composite films and mupirocin alone. By day 19, CBNF-treated wounds achieved 99.21% closure, significantly outperforming the controls that achieved only 83.14% closure (*p* < 0.001). However, the wounds treated with CBF and MDA were neither completely epithelialized nor did the granulation tissues replace the necrotic tissues. A large number of necrotic tissues remained in the untreated tissues. The wound treated with CBF and MDA powder was re-epithelialized. However, fibrosis did not entirely replace granulation tissues, and hair follicles were not completely healed. The rate of wound healing was found to increase in the following order: control < CBF < MDA < CBNF.

## 3. Discussion

CS-based films tend to become brittle after solvent casting; hence, incorporating plasticizers is crucial for enhancing the flexibility of chitosan films. To improve the processability, flexibility, and ductility of the polymeric films, polyethylene glycol (PEG) was selected as a plasticizer.

Wound healing is a dynamic process involving soluble mediators, various cells, and an extracellular matrix. These components are involved in many different processes or steps in healing, including coagulation, inflammation, fibroplasia, collagen deposition, epithelialization, and scar contraction with remodeling [24]. The processes can be organized into three phases: inflammation, fibroplasia, and remodeling [25]. The healing events occur orderly and timely, and there is a significant overlap between each process [26]. In the context of veterinary medicine, chitosan was found to enhance the functions of polymorpho-nuclear leukocytes (PMN) (phagocytosis, and production of osteopontin and leukotriene B4), macrophages (phagocytosis, and production of interleukin^−1^, transforming growth factor b1, and platelet-derived growth factor), and fibroblasts (production of interleukin-8). They observed that chitosan promotes granulation and organization, and therefore, it is beneficial for open wounds; certain PMN functions are enhanced, such as phagocytosis and the production of chemical mediators [27]. A peculiarity of chitosan is its ability to foster adequate granulation tissue formation accompanied by angiogenesis and regular deposition of thin collagen fibers, a property that further enhances the correct repair of dermo–epidermal lesions [28]. The primary biochemical effects of chitin and chitosan are fibroblast activation, cytokine production, giant cell migration, and stimulation of type IV collagen synthesis [29].

Fourier-transform infrared spectroscopy (FTIR) analysis was conducted to investigate the chemical interactions in chitosan (CS), mupirocin (Mup), and polyethylene glycol (PEG) composite films (CBNF). The O―H stretching vibrations (3415 cm^−1^) and amide I band (1648 cm^−1^) from chitosan moved to lower wavenumbers with the blending of PEG and Mup. Additionally, there was a shift in the C―O stretching vibration from the C―O―H of chitosan (1108 cm^−1^). These changes point to a chitosan and PEG intermolecular interaction that is attractive. In the blend spectra, some bands showed minor intensity shifts and position shifts, but no new bands were formed nor did recognizable bands disappear. This suggests a unique interaction, most likely brought about by recently created hydrogen bonds between the chitosan and PEG molecules [23,30]. These results indicate that the composite film was successfully fabricated with CS, PEG, and Mup.

Chitosan-based nanoparticulate films (CBNF) are transparent (Figure 2). The SEM image (Figure 2) reveals a porous, interconnected structure, suggesting high water-retention capacity. It is attributed to the high degree of deacetylation of chitosan (88%), which facilitates hydrogen bonding with water, aiding in the accommodation of both small- and macro-molecules. Okafor and his colleagues developed buccal films made of bio-adhesive polymers: carbopol and Pluronic 127 [31]. SEM micrographs of the buccal films depicted smooth surfaces. Similarly, Üstündağ Okur and his coworkers revealed non-porous and smooth surfaces made of chitosan, polypropylene glycol, and PEG [32]. In contrast, our test samples CBF and CBNF showed porous and interconnected structures. This might be due to the deacetylated and low-molecular-weight form of chitosan [32] derived from fungi.

The swelling ratio of CBF and CBNF, measured by weight change after immersing in PBS, increased significantly from 250% to 1700% over 6 h, stabilizing after 4 h. Both films showed similar water absorption capabilities and moderate water retention over time. The previous report showed evidence that mupirocin-loaded CS/polycaprolactone scaffolds swelled more than polycaprolactone scaffolds [33]. The material absorbing the moisture during the dressing could improve the wound healing ability by releasing the drug molecule. In a study, adhesive mupirocin-loaded CS–glycerine–carbopol composite films released the drug after 1.5 h. Swelling studies indicate that the composite films reach their maximum swelling ratio at 1 h, after which the film’s weight decreases due to the hydrophilic nature of carbopol [32]. The results indicate that both CBF and CBNF had good water absorption and retention ability.

The in vitro release of mupirocin (Mup) from chitosan-based nanoparticulate films (CBNF) was higher at pH 5.4 compared to pH 7.4 over a 72-hour incubation period, with cumulative release percentages of 65% and 20%, respectively. This enhanced release at pH 5.4 is attributed to the protonation of the amine groups in chitosan, facilitating greater release of the Mup in acidic conditions. Amrutiya and colleagues (2009) studied the in vitro release profiles of mupirocin-based formulations using a cellulose dialysis membrane. They compared mupirocin ointment and emulgel by examining their release profiles over 4 and 10 h, respectively. The mupirocin emulgel demonstrated sustained release for up to 24 h. Drug release from the emulgel followed a diffusion-controlled mechanism, with R^2^ values ranging from 0.9738 to 0.9790. The results indicated a diffusion-controlled release pattern and were supported by drug deposition studies [34]. Additionally, mupirocin-loaded chitosan/polycaprolactone scaffolds have been created for wound dressings, and their drug release, measured via UV spectrophotometry, showed a 28.7% release within 24 h and a cumulative 38.1% by 120 h, indicating sustained release. This pattern is attributed to the scaffold’s surface wettability and diffusion rate [25]. Conversely, a study was conducted to prepare double-layer nanofibrous scaffolds. The initial rapid release of mupirocin from these scaffolds resulted in 57% of the total mupirocin content being released within the first 6 h. In comparison, only 30% of the mupirocin diffused from the scaffolds in the same period. This lower release rate was attributed to the strong chemical linkage interactions between PCL and mupirocin conjugated to the fiber, which reduced the release capability of mupirocin [35].

Drug release involves three main kinetic mechanisms: erosion, swelling, and diffusion. Mupirocin (Mup) release from CBNF was studied via various mathematical models, such as zero-order, first-order kinetics, Hixson–Crowell, Higuchi, and Korsmeyer-Peppas. The optimal model was determined by the highest R^2^ value obtained from linear regression analysis. Mup release highly fitted with the Higuchi model—which describes drug release as a diffusion process based on Fick’s law—when it was released from CBNF at 37 °C. This suggests that the primary mechanism controlling Mup’s release from CBNF was through diffusion and polymer degradation. This is in line with research conducted by Pettinelli and colleagues, who created a polyhydroxybutyrate-based hydrogel to encapsulate mupirocin and ketoprofen at the same time. Compared to the individual release from microparticles and hydrogel, the hydrogel demonstrated a slower release for both medications. At 37 °C, the release was seen over seven days. Furthermore, the release pattern was fitted to the Higuchi model, indicating that diffusion was primarily responsible for the release [36]. In conclusion, our findings indicate that the primary mechanism for Mup’s release from CBNF was diffusion through the polymer matrix [37].

Because of their natural origin and capacity for wound healing, chitosan-based wound management products are useful in the treatment of deep wounds and severe burns [22,38]. Although chitosan-based scaffolds have drawbacks like poor mechanical strength and low antibacterial activity, they still promote re-epithelialization and granule layer regeneration, which further improve wound healing [39]. When medications are combined with chitosan-based products, the antibacterial effects are strengthened, and oxidative stress is managed, which speeds up the healing of wounds. Consequently, we suggested that CBNF—a chitosan-based biopolymer mixed with the antibacterial medication mupirocin—would be an ideal option for healing wounds.

Measuring a wound size assesses the amount of tissue damage. This provides baseline information and helps track treatment progress, thus assisting with auditing and predicting treatment efficacy [40]. Wound assessment is essential to effective wound management, and wounds should be measured each time they are assessed [41,42,43]. Recording the wound area and volume is a routine part of animal assessment and provides information on the healing progress. The practice of using clinical photography is common, and measuring the size of wounds is simple and easy. There have been a few studies where Image J software (version 1.54) has been used to measure wounds [43,44,45]. Hence, the present study applied digital photography technology to the wound image data acquisition, and computer-assisted flat-area calculation software and area under the measurements were used. By day 19, CBNF-treated wounds achieved 99.21% closure, significantly outperforming the controls, which achieved only 83.14% closure (*p* < 0.001).

In the present study, histological findings on wounded skin dressed in CBNF indicate that collagen fibers were fine in the wounds and more mature than in the control (7 days post-treatment); their arrangement was closely similar to that in normal skin. On the 19th day, it was observed that the wounds were completely re-epithelialized, granulation tissues were almost replaced with fibrosis, and hair follicles were almost healed. The mode of action of wound healing [46] with chitin-based dressings was depolymerized and further hydrolyzed to N-acetyl glucosamine, evidenced by incorporation into glycoproteins.

Re-epithelialization is crucial in wound healing to restore the skin barrier and reduce inflammation. In this study, the wound demonstrated complete re-epithelialization and well-aligned collagen, indicating effective healing. Similarly, Li and his colleagues developed a wound-healing application and loaded it with a curcumin nanoformulation in MPEG–chitosan (methoxy polyethylene glycol graft chitosan) films [47]. CBNF was completely re-epithelialized (yellow arrow), granulation tissues were nearly replaced by fibrosis, and hair follicles were almost healed, indicating that the combined usage of chitosan, PEG, and the mupirocin nanoformulation could significantly accelerate the re-epithelialization of wounds (Figure 6 and Figure 7).

In another study, a biocompatible chitosan/polyethylene glycol diacrylate (PEGDA) blend films were prepared by the Michael addition reaction (nucleophilic addition of a carbanion or another nucleophile to an α, β-unsaturated carbonyl compound) with different weight ratios as wound-dressing materials [44]. The mechanical and swelling properties of chitosan were found to be enhanced after the chemical modification. Indirect cytotoxicity assessment of films with mouse fibroblasts (L929) indicated that the material showed no cytotoxicity towards the growth of L929 cells and had good in vitro biocompatibility [48]. HemCon^®^ bandage is an engineered chitosan acetate preparation designed as a hemostatic dressing and is under investigation as a topical antimicrobial dressing [49]. The conflicting clamping and stimulating effects of the chitosan acetate bandage on normal wounds were studied by removing the bandage from wounds at times after application ranging from 1 h to 9 days. The results showed that the 19th day observation gave the earliest wound closure, and all application times gave a faster healing slope than control wounds. It was also observed that the chitosan acetate bandage reduced the number of inflammatory cells in the wound on the second and fourth days and had an overall beneficial effect on wound healing, especially during the early period [49]. Hence, our histological analysis revealed that wounds treated with CBNF showed no specific inflammation, reactive granulomas, or microorganisms, and exhibited more significant collagen deposition and well-aligned collagen fibers by day 19 compared to the CBF and MDA treatments. These findings indicate superior wound-healing properties of CBNF, as evidenced by the enhanced collagen organization and the absence of adverse reactions. The effectiveness of wound healing was noted in the following order: control wounds showed the most negligible improvement, followed by wounds treated with CBF, then MDA, and most effectively, CBNF. Thus, the obtained results clearly showed the superiority of this CBNF over the other wound-healing agents.

The topically administrated curcumin–MPEG–chitosan film in rats resulted in faster wound reduction and healing than when treated with MPEG–chitosan film. Furthermore, they showed that re-epithelialization was faster in the curcumin–MPEG–chitosan film-treated wounds. Masson’s trichome staining indicated that the curcumin–MPEG–chitosan film wound had compact and well-aligned collagen, whereas collagen was loose and had an irregular arrangement in the MPEG–chitosan film wound. However, they mentioned no evidence of complete re-epithelialization from Masson’s trichome or H&E staining methods [47]. In contrast to the present study, re-epithelialization of the wound was observed in the treated group (Figure 6) on the 19th day of post-wounding in the experimental animals.

Moreover, defective wound epithelialization was seen in CBF- and MDA-treated rats, following Masson’s trichome staining method. The superior biodegradability and hydrophilicity of CBNF could likely enhance its compatibility with wounded tissues and increase its activity as a wound-healing accelerator. In the case of CBNF, the interaction between the wounded site and the healing agent was maximized, which resulted in the highest strength of the treated skin; this may be due to the enhanced re-epithelialization and granulation of tissues. Consequently, the CNBF is considered an ideal biomaterial with antibacterial, wound-healing properties and easy application, suggesting its potential to develop into a new effective wound-healing agent.

## 4. Materials and Methods

### 4.1. Materials

Mupirocin was purchased from Himedia, Mumbai. Chitosan was extracted from the test fungus. Glacial acetic acid was purchased from Sigma Chemical Co. (St Louis, MO, USA), and polyethylene glycol (PEG) and dimethyl sulfoxide (DMSO) from Merck (Darmstadt, Germany). All other reagents and solvents used were of pharmaceutical grade. The statistical analyses were conducted utilizing GraphPad Prism (GraphPad 10.0, San Diego, California, USA) software.

#### 4.1.1. Preparation of Chitosan-Based Nanoparticulate Films (CBNF)

Chitosan-based nanoparticulate films (CBNF) were successfully developed and prepared using the modified casting/solvent evaporation method. Chitosan was dissolved in acetic acid (1%) containing mupirocin soluble in dimethyl sulfoxide (2 mL) and stirred overnight, and then polyethylene glycol (2%) was added before mupirocin as a plasticizer. The resulting solution was sonicated to remove air bubbles, dropped into a Petri dish (10 mL), and then allowed to dry at room temperature. After drying, the films were peeled off and stored in an airtight container at room temperature until further investigation.

#### 4.1.2. Characterization of the Test Samples

The composite films were characterized by Fourier-transformed infrared spectroscopy (FTIR) by the KBR disc method.

The test samples were mounted on metal grids with double-sided adhesive tape, coated with a gold sputter coater (Quorum Technologies, Newhaven, UK) under high vacuum, 0.1 Torr, 1.2 kV, and 50 mA at 25 °C ± 1 °C. The surface morphology of coated samples was examined by field emission scanning electron microscopy (FESEM; HITACHI, Tokyo, Japan) at 15 kV. Dynamic Light Scattering (DLS) at a refractive index of 1.33 and an operating wavelength of 632 nm was used to calculate the hydrodynamic diameters of the CBN solution. The test sample was filtered with a 0.02 nm syringe filter and immersed in particle-free water, followed by ultrasonication in an ultrasonic bath for 15 min.

#### 4.1.3. Determination of Swelling and Moisture Retention 

The swelling and moisture retention abilities of CBF and CBNF were evaluated at various time intervals from 0 to 6 h in deionized water. In brief, the film was cut into 2 × 2 cm and the pre-weighed (W_0_) film immersed in 5 mL of deionized water. Then, at each time interval, the wet film was taken out, the excess water was gently rubbed with filter paper, and the film was weighed (W_n=1-6_). The percentage of swelling is determined according to the following formula.
(1)$$\%\;of\;swelling=(Wn-W_{0})/W_{0}\times 100$$

The moisture retention ability was determined to be inversely proportional to the swelling ability of the film. In brief, the dry film was immersed in deionized water for 2 h. Then, the initial weight of the wet film was taken and it was kept at room temperature on a glass plate. The weight of the film was recorded for each predetermined time interval until its dry weight. To determine the mean value, the experiments were run in triplicate.

#### 4.1.4. In Vitro Drug Release

The release rate of Mup from the CS-PEG-Mup film at different pHs (5.4 and 7.4) was determined by in vitro dissolution method. In brief, 20 mg of the film was added in different pH-containing buffer solutions, transferred to the preactivated dialysis membrane, and immersed into the 100 mL dissolution bath. At predetermined intervals, 5 mL of the samples were withdrawn from the dissolution bath and the same volume of respective buffer solution. The Mup concentration of the sample collected from each time interval was assessed by the known concentration of Mup at λ_Max_ 220 nm. The cumulative Mup release percentage was calculated. To determine the mean value, the experiments were run in triplicate. In addition, the mupirocin (Mup) release from CBNF was fitted with various mathematical drug release models, such as zero-order, first-order kinetics, Hixson–Crowell, Higuchi, and Korsmeyer-Peppas.

#### 4.1.5. Evaluation of Antibacterial Activity

The antibacterial activity of CBCF and CBNF against Gram-positive (*Staphylococcus aureus MTCC 740*) and Gram-negative (*Pseudomonas aeruginosa MTCC 741*) bacteria was evaluated using the disk diffusion method and compared with the antibiotic mupirocin. Six-millimeter-diameter discs of the test samples (Mup, CBF, and CBNF) were placed on the surface of the inoculated plates. Test samples were placed on pathogen-inoculated plates and incubated at 37 °C for 24 h, and the resulting inhibition zones were measured in millimeters. To determine the average, the experiments were carried out in triplicate.

#### 4.1.6. Animal Studies

The animal study was performed with male Wistar rats (8–10 weeks) weighing between 150 g and 170 g at the time of the experiments. The Ethics Committee approved the animal protocols followed in the present study for the University of Madras, and according to the guidelines, the animals were maintained under laboratory conditions. Rats were individually anesthetized via an IP injection (40 mg/kg ketamine, 5 mg/kg xylazine) to create the wound. Before that operation, the skin area was shaved and disinfected using ETOH. Then, the animal’s dorsal skin was removed using a sterilized surgical knife. Wounds of 2 cm diameter were created with no visible bleeding. The animals were divided into IV groups. Group I was considered as a control and was not treated; group II was treated with chitosan film (CBF); group III was treated with mupirocin alone (MDA); group IV was treated with chitosan film containing mupirocin (CBNF) (Table 2). Animals were kept in separate cages and were fed commercial rat food and water. All animals showed good general health conditions throughout the study, as assessed by their weight gain. The biopsy of samples was taken on the 19^th^ day from each rat, and the degree of healing was evaluated by histopathological analysis. To determine the mean value, the experiments were run in triplicate.

#### 4.1.7. Evaluation of Wound Healing

The wounds were photographed after placing a ruler (scale) next to the wound. These pictures were then analyzed using Image J^TM^. The edges of the wound were marked, and the number of pixels falling under the ruler technique (maximum length and width of the wound) and the marked wounds were calculated. Utilizing the known dimensions of the ruler (scale), the precise size of the marked wound area could be determined, as described in [40]. During the measurement procedure, the first image was displayed on the computer screen, and then the horizontal dimension using the embedded calibration bar or scale.

#### 4.1.8. Calculation of Percent Area Reduced (PAR)

Percent area reduced (PAR) was calculated following Cardinal et al.’s method [50], as the difference in wound area (%) between consecutive study visits from days 0 to 19 for each animal was noncumulative. Average PAR was computed as the overall mean percent reduction over time per animal. To determine the mean value, the experiments were run in triplicate. A two-way ANOVA analysis was used for multiple comparisons.
(2)$$PAR=\frac{{Area}_{0}-{Area}_{i}}{{Area}_{0}}\times 100;i=1,2,3,4,5,\ldots \;\ldots$$

#### 4.1.9. Histological Analysis

The material from the skin lesions obtained by necropsy was formalin-fixed and paraffin-embedded for routine histological processing. A 3-millimeter section obtained from each paraffin block was stained with hematoxylin and eosin (H&E) and evaluated in a blinded manner by two observers using a light microscope with specific image analysis software from Olympus (Olympus Stream 2.4). For the morphological evaluation of skin lesions, the following parameters were considered: wound bed length, the granulation tissue layer thickness, and the epithelial layer thickness. In assessing these three parameters, morphology was always considered the most significant dimension observed. Skin fragments with no chitosan film were used as normal controls. To determine the mean value, the experiments were run in triplicate.

## 5. Conclusions

This study using chitosan-based nanoparticulate films (CBNF) showcased enhanced wound healing, with CBNF-treated wounds displaying rapid closure rates, reduced wound areas, and superior collagen deposition compared to controls and other treatments. Macroscopic and histological assessments revealed CBNF’s effectiveness in promoting re-epithelialization and collagen alignment by day 19. CBNF’s biocompatibility and accelerated wound closure, reaching 99.21% closure by day 19, highlight its potential as a wound-healing agent. This biomaterial’s antibacterial property and ease of application position CBNF as a ideal candidate for further development in wound care.

## Data Availability

The data presented in this study are available on request from the corresponding author.

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
