# Peer review of "Fabrication of Mupirocin-Loaded PEGylated Chitosan Nanoparticulate Films for Enhanced Wound Healing"

_ijms, 2024, doi:10.3390/ijms25179188_

Round 1

Reviewer 1 Report

Comments and Suggestions for Authors

In this manuscript, the authors have prepared and characterized a mupirocin (Mup)-loaded PEGylated chitosan (CS-PEG) nanoparticulate film (NF)[CBNF]. The film showed high water absorption (≥1700%) and moderate water retention ability within 6h. Histological analysis revealed significant development, with re-epithelialization and granulation of tissues after 19 days, indicating the healing efficiency of CNBF, suggesting that it could be an effective carrier and delivery agent for Mup-like anti-inflammatory drugs. However, several major questions need to be addressed.

1.      The sentence: “Numerous findings indicate the beneficial use of chitosan in gels, films, and meshes for treating skin wounds, demonstrating positive outcomes in both human and animal studies" (lines 73-75) is redundant. What are the positive outcomes? Also, the sentence should have adequate citations to support these claims.

2.      The authors mention at the beginning of the results section text that the films contain “mupirocin-loaded chitosan nanoparticles”. The synthesis of these nanoparticles, their dimensions, and their properties (e.g., zeta potential) is nowhere in the manuscript.

3.      Throughout the text the authors specify that “By cross observation morphologically chitosan-based nanoparticulate films (CBNF) were transparent and the film was thinly layered” (lines 133-134) and “SEM analysis of chitosan-based nanoparticulate films (CBNF) shows that these films are transparent and thinly layered” (lines 303-304). The technique in question (FESEM) does not allow to infer the optical transparency of the films. The authors should measure the optical transmittance of the films and provide macroscopic images of the materials, showing their transparency. Also, the thickness of the films should be measured and presented.

4.      Several plots in the manuscript do not contain error bars (see Figures 3 and 5). Also, the number of replicates (n) for each analysis should be stated in the legends (and in the materials & methods section, as discussed below).

5.      Figure 4 on the “Macroscopic wound healing with different treatments” should include the titles above each column of images, instead of the designations (a) through (d), to facilitate interpretation.

6.      The authors compare the drug release results with similar studies (lines 326-347) wherein the drug release mechanism is addressed, but no calculation was performed for this work. The authors should fit the release data to find the appropriate release mechanism and include that information in the results and discussion.

7.      The acronyms present in the legend of Figure 6 are not the same as those presented in the figure.

8.      The authors suggest that “CNBF is considered to be one of the ideal biomaterials with biocompatibility, biodegradability, and wound-healing properties…” (lines 422-423), but it is not clear that the performed biocompatibility studies support their claims. This should be better evaluated (e.g., wound healing assays) and discussed within the text.

9.      The materials and methods sections should be improved to include the n for each analysis/experiment.

10. How was the statistical analysis performed? This should be specified in the article to verify the adequacy of the data interpretation. 

11. Authors need to revise the references so that they are all in the same format (capitalization, revise doi's). 

Comments on the Quality of English Language

The authors need to revise the language and general phrasing of the document. Several sentences within the text are confusing and poorly written, particularly in the introduction (see lines 83-91, the sentence is incoherent).

Author Response

Reviewer #1: Manuscript ID: ijms-3119820

Title: “Fabrication of Mupirocin-Loaded PEGylated Chitosan Nanoparticulate Films for Enhanced Wound Healing”

In this manuscript, the authors have prepared and characterized a mupirocin (Mup)-loaded PEGylated chitosan (CS-PEG) nanoparticulate film (NF)[CBNF]. The film showed high water absorption (≥1700%) and moderate water retention ability within 6h. Histological analysis revealed significant development, with re-epithelialization and granulation of tissues after 19 days, indicating the healing efficiency of CNBF, suggesting that it could be an effective carrier and delivery agent for Mup-like anti-inflammatory drugs. However, several major questions need to be addressed.

Comment

  1. The sentence: “Numerous findings indicate the beneficial use of chitosan in gels, films, and meshes for treating skin wounds, demonstrating positive outcomes in both human and animal studies" (lines 73-75) is redundant. What are the positive outcomes? Also, the sentence should have adequate citations to support these claims.

Response

                Citations have been included following the request of the reviewer.

Comment

  1. The authors mention at the beginning of the results section text that the films contain “mupirocin-loaded chitosan nanoparticles”. The synthesis of these nanoparticles, their dimensions, and their properties (e.g., zeta potential) is nowhere in the manuscript.

Response

Mupirocin-loaded chitosan nanoparticles were examined through Electron Microscopy. The size of these nanoparticles was mentioned in the revised manuscript using the DLS method.

Comment

  1. Throughout the text the authors specify that “By cross observation morphologically chitosan-based nanoparticulate films (CBNF) were transparent and the film was thinly layered” (lines 133-134) and “SEM analysis of chitosan-based nanoparticulate films (CBNF) shows that these films are transparent and thinly layered” (lines 303-304). The technique in question (FESEM) does not allow to infer the optical transparency of the films. The authors should measure the optical transmittance of the films and provide macroscopic images of the materials, showing their transparency. Also, the thickness of the films should be measured and presented.

Response

Thank you very much for your constructive information. Yes, FESEM doesn’t depict the transparent morphology of the film (the sentence has been corrected). In Fig. 3d the photographic image is depicted. Since this manuscript has to be submitted within the stipulated date, your valuable suggestion regarding optical transmittance and measurement of thickness shall be considered in future directions.

Comment

  1. Several plots in the manuscript do not contain error bars (see Figures 3 and 5). Also, the number of replicates (n) for each analysis should be stated in the legends (and in the materials & methods section, as discussed below).

Response

As suggested by the reviewer, the images (Fig.3 and Fig. 5) with error bars have been replaced. All the experiments were conducted in triplicates, which has been mentioned in the legends.

Comment

  1. Figure 4 on the “Macroscopic wound healing with different treatments” should include the titles above each column of images, instead of the designations (a) through (d), to facilitate interpretation.

Response

Fig. 5 (Modified Fig. 4) has been modified following the request by the reviewer.

Comment

  1. The authors compare the drug release results with similar studies (lines 326-347) wherein the drug release mechanism is addressed, but no calculation was performed for this work. The authors should fit the release data to find the appropriate release mechanism and include that information in the results and discussion.

Response

Thank you so much for your valuable comments. The in vitro release of Mup form CBNF at different pH was fitted with various mathematical drug release models and represented in the revised version.

Comment

  1. The acronyms present in the legend of Figure 6 are not the same as those presented in the figure.

Response

In Fig.8 (Modified Fig. 6) The legend has been corrected in the legend as mentioned by the reviewer.

Comment

  1. The authors suggest that “CNBF is considered to be one of the ideal biomaterials with biocompatibility, biodegradability, and wound-healing properties…” (lines 422-423), but it is not clear that the performed biocompatibility studies support their claims. This should be better evaluated (e.g., wound healing assays) and discussed within the text.

Response

Thank you for your constructive review, the sentences have been modified and replaced accordingly.

Comment

  1. The materials and methods sections should be improved to include the n for each analysis/experiment.

Response

The n for the respective analysis is inserted in the respective experimental sections.

Comment

  1. How was the statistical analysis performed? This should be specified in the article to verify the adequacy of the data interpretation.

Response

Statistical analyses were conducted utilizing GraphPad Prism (GraphPad 10.0, USA) software. To analyze the significant differences between groups, Two-way ANOVA analysis is used for multiple comparisons, with a p< 0.05 being deemed statistically significant.

Comment

  1. Authors need to revise the references so that they are all in the same format (capitalization, revise doi's).

Response

All the Captions and corrections in the references are updated.

Comments on the Quality of English Language

The authors need to revise the language and general phrasing of the document. Several sentences within the text are confusing and poorly written, particularly in the introduction (see lines 83-91, the sentence is incoherent).

Response

The language and general wording of the document have been paraphrased throughout the manuscript. Incoherent sentences were checked and reworded as requested by the reviewer.

Comments on the Quality of English Language

Minor editing of English language required.

Language editing and general wording of the document have been paraphrased throughout the manuscript. 

Reviewer 2 Report

Comments and Suggestions for Authors

Azeez et al. have conducted an insightful study on the preparation and characterization of a mupirocin-loaded PEGylated chitosan nanoparticulate film (CBNF) aimed at enhancing skin regeneration and wound healing. Their work demonstrates the successful incorporation of nanoparticles into the chitosan matrix, as confirmed by FTIR spectroscopy and SEM analysis, and highlights the film's significant water absorption and retention properties, alongside its promising histological outcomes indicating effective wound healing.

After careful consideration, I have decided to recommend major revisions for this manuscript. Addressing the necessary improvements will enhance the clarity, depth, and overall impact of the study.

1、The authors should discuss the advantages of their study in comparison to other tissue repair materials in the introduction (https://doi.org/10.1039/D3CS00923H), as well as the advantages of the antimicrobial methods used in the current study (https://doi.org/10.1016/j.mtbio.2023.100582, https://doi.org/10.1007/s12274-022-5129-1). This will provide a clearer context and highlight the significance of their work in the field.

2、In the infrared spectra, please mark the characteristic peaks and provide a detailed analysis. This will help in clearly identifying the functional groups present and their modifications. For example, the changes related to hydrogen bonding are not apparent in the infrared spectra provided.

3、The authors claim a specific reaction mechanism; please provide a reaction mechanism diagram to support this claim. This will make the proposed mechanism more comprehensible and credible.

4、At the same 5-micron scale, why are the nanoparticles not observable in Figure 2a? Clarification is needed to understand the distribution and visibility of nanoparticles within the film matrix.

5、The release of mupirocin data appears to lack error bars. Did the authors only conduct a single experiment? Including error bars and mentioning the number of replicates will strengthen the reliability and reproducibility of the results.

6、Please include relevant antibacterial tests, such as supplementing with inhibition zone tests. This will provide a more comprehensive evaluation of the antibacterial efficacy of the film.

7、Simply measuring the change in wound size is insufficient. Quantitative characterization should also include collagen deposition and hair follicle counts in stained tissue sections. This will provide a more detailed assessment of the healing process and the effectiveness of the treatment.

8、The manuscript includes extensive discussion on the swelling and moisturizing capacity of the film. However, it is unclear whether the authors have tested the film’s ability to absorb exudate and retain moisture during skin wound healing tests. Including these assessments would provide a more comprehensive evaluation of the film’s effectiveness in wound healing applications.

Comments on the Quality of English Language

Minor editing of English language required.

Author Response

Reviewer – 2

Comments and Suggestions for Authors

Azeez et al. have conducted an insightful study on the preparation and characterization of a mupirocin-loaded PEGylated chitosan nanoparticulate film (CBNF) aimed at enhancing skin regeneration and wound healing. Their work demonstrates the successful incorporation of nanoparticles into the chitosan matrix, as confirmed by FTIR spectroscopy and SEM analysis, and highlights the film's significant water absorption and retention properties, alongside its promising histological outcomes indicating effective wound healing.

After careful consideration, I have decided to recommend major revisions for this manuscript. Addressing the necessary improvements will enhance the clarity, depth, and overall impact of the study.

Comment

  1. The authors should discuss the advantages of their study in comparison to other tissue repair materials in the introduction (https://doi.org/10.1039/D3CS00923H), as well as the advantages of the antimicrobial methods used in the current study (https://doi.org/10.1016/j.mtbio.2023.100582, https://doi.org/10.1007/s12274-022-5129-1). This will provide a clearer context and highlight the significance of their work in the field.

Response

The introduction of the study has been modified as referred by the reviewer and references are included in the revised manuscript.

Comment

  1. In the infrared spectra, please mark the characteristic peaks and provide a detailed analysis. This will help in clearly identifying the functional groups present and their modifications. For example, the changes related to hydrogen bonding are not apparent in the infrared spectra provided.

Response

             Based upon valuable suggestions by the reviewer, Fig.1 has been labeled with characteristic peaks in FTIR spectra. A comprehensive analysis has been provided in the revised version.

Comment

  1. The authors claim a specific reaction mechanism; please provide a reaction mechanism diagram to support this claim. This will make the proposed mechanism more comprehensible and credible.

Response

Schematic representation has been updated in the manuscript as Scheme 1.

Comment

  1. At the same 5-micron scale, why are the nanoparticles not observable in Figure 2a? Clarification is needed to understand the distribution and visibility of nanoparticles within the film matrix.

Response

Fig. 2a shows the morphological properties of CPCF composite films in which nanoparticles are not formed because they are polymer blends in which cross-linking has not occurred resulting in non-particulate films.

Comment

  1. The release of mupirocin data appears to lack error bars. Did the authors only conduct a single experiment? Including error bars and mentioning the number of replicates will strengthen the reliability and reproducibility of the results.

Response

The error bars for the mupirocin release study are included in the manuscript (Fig.3c). The experiments mentioned were carried out in triplicate.

Comment

  1. Please include relevant antibacterial tests, such as supplementing with inhibition zone tests. This will provide a more comprehensive evaluation of the antibacterial efficacy of the film.

Response

Based on your suggestions, Antibacterial studies against Gram-positive and Gram-negative bacteria have been carried out and included in the study.

Comment

  1. Simply measuring the change in wound size is insufficient. Quantitative characterization should also include collagen deposition and hair follicle counts in stained tissue sections. This will provide a more detailed assessment of the healing process and the effectiveness of the treatment.

Response

We appreciate your suggestion for a more thorough evaluation of wound healing. Due to the research work already being concluded, we are unable to incorporate these assessments at this time, but we will certainly consider them in future studies to enhance the evaluation of treatment effectiveness.

Comment

  1. The manuscript includes extensive discussion on the swelling and moisturizing capacity of the film. However, it is unclear whether the authors have tested the film’s ability to absorb exudate and retain moisture during skin wound healing tests. Including these assessments. would provide a more comprehensive evaluation of the film’s effectiveness in wound healing applications.

Response

We acknowledge the insightful suggestion and the importance of testing the film's ability to absorb exudate and retain moisture during skin wound healing, and we agree that these assessments would greatly enhance our study. However, due to unavoidable circumstances, we are unable to conduct these tests at this time. We will certainly consider including them in future studies.

Round 2

Reviewer 1 Report

Comments and Suggestions for Authors

The authors addressed most of the reviewer's comments; Thus, the manuscript could be accepted in the present form.

Reviewer 2 Report

Comments and Suggestions for Authors

Thanks for the author's response, I have no further questions at this time and recommend acceptance of the paper.